# Properties of Elasto-Hydrodynamic Oil Film in Meshing of Harmonic Drive Gears

**DOI:** 10.3390/ma14051194

**Published:** 2021-03-03

**Authors:** Adam Kalina, Aleksander Mazurkow, Waldemar Witkowski, Bartłomiej Wierzba, Mariusz Oleksy

**Affiliations:** 1Faculty of Mechanical Engineering and Aeronautics, Rzeszów University of Technology, 35-959 Rzeszów, Poland; almaz@prz.edu.pl (A.M.); wwitkowski@prz.edu.pl (W.W.); bwierzba@prz.edu.pl (B.W.); 2Faculty of Chemistry, Rzeszów University of Technology, 35-959 Rzeszów, Poland; molek@prz.edu.pl

**Keywords:** harmonic drive, geometry, mesh, wave generator, rigid circular spline, flexspline, oil film, elasto-hydrodynamic oil film, EHD, mechanical engineering

## Abstract

Among the essential issues facing designers of strain wave gears, the provision for correct lubrication should be of paramount importance. The present paper presents the results of research on elasto-hydrodynamic oil film in meshing of a harmonic drive with an involute tooth profile. The research was carried out based on theoretical models developed by Dowson and Higginson. For the discussed structural problem, results of the study are presented graphically in the form of static characteristics of the oil film. Correct operation regimes were determined for two different oils. The paper also provides a review of information concerning the design and principle of operation of strain wave transmission.

## 1. Introduction

### 1.1. Modern Drive System Design Solutions

Dynamic development of technology has resulted in the automation and robotization of not only manufacturing processes, but also many aspects of everyday life. A good example here is a device for neurorehabilitation, the prototype of which together with the results of its experimental examination was presented by Yamine et al. [1]. The device, equipped with two drives and transmissions providing a reduction rate of 49:1, was designated to support the process of rehabilitation in patients with physical motor disabilities, which made it difficult for them to carry out everyday activities. It is also worth mentioning other devices intended for operation in outer space or on other planets [2].

In view of the above, it is advisable to improve and adapt existing transmission designs and develop new ones, capable of meeting the sophisticated requirements demanded from modern drive systems. Such requirements include, among other things: high accuracy and precision in the transfer of motion; the possibility to obtain large transmission ratios at small overall dimensions of the gear; high efficiency; operability in extreme conditions (e.g., low temperatures [3]); and the overall high culture of drive train operation. The following may serve as examples of research projects aimed at the development of existing solutions:—The use of duplex worm gears with an adjustable backlash in the joints of robot manipulator arms [4]. The paper presents the results of research on the improvement of motion transfer precision in the drive and minimization of vibrations which were obtained as a result of the reduction in play in meshing by means of the use of a special housing;—Modal analysis of a planetary transmission gear [5]. The paper quotes results of numerical analyses carried out in the ANSYS Workbench environment aimed at the determination of frequencies and forms of natural vibrations of components making up a planetary transmission.

One of the newest design solutions in the area of transmissions for modern drive systems is the strain wave transmission, also known as the harmonic drive. The design, in view of its exceptional properties, is used in, among other things, drive trains of devices intended for operation in outer space [6]. The design solution was first patented in 1959 in the United States by C.W. Musser [7,8,9,10].

Among the main merits of harmonic drives, the following are most important:Smooth and silent operation which is a result of multiple-pair intermeshing where up to 50 pairs of teeth may be engaged at the same time;Precise transmission of motion due to virtually no backlash in meshing;Large reduction ratios available in a single step, up to the value of *i_r_* = 350;High transmission efficiency;Small overall dimensions and compactness of the structure;A wide range of design variants and configurations.Harmonic drives have also their flaws, which include, for instance:The risk of occurrence of fatigue damage as a result of elastic deformations occurring in the thin-walled flexspline;The possibility of occurrence of the teeth profiles interference effect, because the gears have small modules with a small backlash in the meshing.

The subject of the present study is a harmonic drive with a double-wave elliptical cam generator. For the assumed input quantities such as the load and contact geometry, the theoretical model proposed by Dowson and Higginson was used to determine properties of the elasto-hydrodynamic oil film occurring in meshing of toothed wheels. The results presented in this paper were obtained as part of a wider research project concerning the properties of harmonic drives.

### 1.2. The Harmonic Drive Design and Principle of Operation

Figure 1 schematically presents the engagement of the main components of a double-wave strain transmission, i.e., the rigid circular spline, the flexspline, and the cam wave generator. More information on the design and the principle of operation of the drive can be found in articles written by Harmonic Drive LLC, Mijał and Ostapski [2,11,12].

In the discussed design solution, the output shaft rotation sense will be opposite to that of the input shaft. On the input shaft, a double-wave generator is fixed. In the case of a strain wave gear with double-wave generator, a single full turn of the input shaft will result in displacement of the rigid circular spline by two tooth spaces. The number of waves, denoted *z_w_*, is the number of engagement areas and is an assumed number, based on which the number the number of flexspline teeth *z*_1_ and circular spline teeth *z*_2_ is determined:(1)$$z_{w}\;=\;z_{1}\;-\;z_{2}$$

The drive reduction ratio *i_r_* is calculated from the relationship:(2)$$i_{r}=-\frac{z_{1}}{z_{w}}=-\frac{z_{1}}{z_{2}-z_{1}}=-\frac{z_{1}}{2}$$

Trajectories of the displacement of points representing the position of a flexspline tooth axis relative to the tooth space of the circular spline are presented in Figure 2. The trajectories depend on, among other things, the design of the generator.

The effect of selected harmonic drive parameters on the form of the trajectories discussed above is discussed in the studies reported in articles written by Mijał, Ostapski, Kalina et al. [11,12,13,14].

### 1.3. Harmonic Drive Lubrication Methods

Strain wave gears can be lubricated with plastic grease or with oil. These two lubrication methods are used typically in applications of drive to manipulators and robots. Table 1 presents a summary of information concerning the grease application recommended by some manufacturers of strain wave gears (Harmonic Drive^®^ (Limburg an der Lahn, German) [2] and Laifual Drive (Zhejiang, China) [15]).

The compactness of the structure characterizing strain wave gears is an argument in favor of the use of plastic grease. Limited space inside the transmission and internally meshed splines allows introduction of the lubricant directly into the tooth spaces of transmission gears. Tooth space bottom function as lubrication pockets and the smallness of space creates favorable conditions for achieving proper conditions of lubrication. Moreover, grease is applied inside the flexspline body (lubrication of surfaces engaged with the outer race of the flexspline bearing) and onto non-working surfaces of the flexspline sleeve (countering of corrosion of abutment surfaces). The plastic grease quantity and application method depend on the conditions under which the drive is operated [2]. Table 2 shows the relationship between the maximum admissible input shaft speed and the lubricant used to lubricate series CFS harmonic drives [2].

It follows from the information contained in Table 2 that the drives lubricated with oil can be operated at higher input shaft speeds compared to gears lubricated with the use of plastic grease. Transmissions lubricated with grease operate in the mixed friction regime, whereas those in which oils are used as a lubricant operate in the fluid friction area. The effect of the lubricant on the coefficient of friction is shown in Figure 3 [16].

In the meshing, the lowest values of the coefficient of friction (Figure 3) are obtained in fluid friction conditions. Properties of the oil film depend significantly on the type of the oil used. An additional advantage of lubrication of that type is the possibility to carry away the heat and contaminants from the gear engagement area together with the flowing oil. Table 3 lists some of the oils intended for the lubrication of harmonic drives [2].

### 1.4. Mathematical Models Used to Describe the Oil Film

The point of departure for many contemporary research projects on the issue of hydrodynamic (HD) and elasto-hydrodynamic (EHD) lubrication in gear transmissions is a study carried out by a team led by Dowson [17]. The presented model was repeatedly modified in view of its usefulness in the description of lubrication of, among other things, roller bearings and plain bearings [18,19]. Within the framework of further research, models intended for the description of lubrication conditions in the engagement area have been developed, which were also based on the results published by Dowson et al. [17]. Examples of such studies include publications concerning, among other things, lubrication in transmission gears:—With an involute teeth profile [20,21];—Bevel gears [22,23];—Hypoid gears [21,24];—Spiroid gears [25];—Worm gears [26,27,28,29].

Available in the literature are studies concerning the use of oils to lubricate strain wave transmissions which, however contrary to the references quoted above, do not present any models capable of describing the properties of the oil film. This follows from, among other things, the difficulties which are posed by the description and experimental verification of the results obtained from theoretical models. Moreover, the harmonic drives, in view of their specific structure, typically have a very small inner space which hinders direct observation of phenomena occurring in the course of engagement of toothed rims. It is also worthwhile mentioning that the design of strain wave gears and factors connected to the specificity of their application (drive systems for robots) favors the selection of plastic greases as lubricants. In view of the above, studies concerning lubrication in harmonic drive gears concern:
—Research on properties of lubricants intended for strain wave gears. Special attention is deserved here to the studies concerning the lubrication of harmonic drive gears designed for operation in outer space [30,31];—The tear and wear of the transmissions and selected tribological aspects [32,33].

Results of theoretical analysis indicate, therefore, that there are good reasons to carry out research aimed at the development of mathematical models capable of describing the parameters of the oil film generated in the area of engagement of harmonic drive gears. This paper presents the algorithm used in the authors’ original model, allowing description of the properties of the oil film in the meshing of a strain wave gear together with results of analysis of the effect of input shaft speed on the oil film minimum height at the selected transmission operating point.

## 2. Materials and Methods

### 2.1. Assumptions and Boundary Conditions

Main assumptions adopted for the developed physical and mathematical model:The problem was considered in planar reference systems;To describe the geometry and kinematics of meshing, two reference systems were adopted, namely *XOY* (with the origin situated on the generator shaft axis) and *X*_1_*O*_1_*Y*_1_ (the origin of which is translated in the direction of the *Y*-axis by the value of *R_nu_* equaling the length of the radius of the neutral layer of non-deformed flexspline). The systems *XOY* and *X*_1_*O*_1_*Y*_1_ are depicted in Figure 4.

The immovable rigid circular spline was oriented in the reference systems *XOY* and *X*_1_*O*_1_*Y*_1_ in such way that the rigid spline coincided with the direction of axes *Y* and *Y*_1_;The flexspline tooth position as a function of the generator rotation angle *φ_G_* was determined in terms of two characteristic points *M* and *N* which define the position of the flexspline tooth axis. Point *M* is situated at the intersection of the flexspline tooth axis with the addendum circle, whereas *N* is situated at the intersection of the tooth axis with the root circle;In view of the symmetry and cyclic nature of the curve describing the generator cam profile shape, the research was carried out for the generator position angle *φ_G_* varying in the range *φ_G_* ϵ < 0,*π*/2 >; Trajectories of points *M* and *N* are symmetrical relative to the tooth space axis;The oil viscosity function and the oil density function take into account the effect of pressure and temperature;The oil flow in the engagement area is consistent with the peripheral direction, whereas the oil flow in both axial and radial directions is negligibly small;The heat is carried away from the contact zone by the engaged surfaces of toothed wheels and/or the flowing oil;The contact surfaces are considered perfectly cylindrical and smooth;At boundary surfaces, the slip effect does not occur—velocity of the oil boundary layer and of the tooth surface was the same;Linear contact of teeth was assumed on the whole width of the toothed rims;The engagement areas are symmetrical and equally loaded;Calculations were carried out for the steady-state operation of the gear transmission;Thermal expansion of toothed rims has no effect on the shape of the flexspline tooth relative path;The model does not take into account the effect of oil anti-wear additives;For each generator position angle *φ_G_* at which the engagement of toothed rims occurs, a substitution model of the contact is constructed consistent with Figure 5.

### 2.2. Algorithm of the Developed Method

In order to determine the properties of the oil film in the gear engagement area of a harmonic transmission it is necessary, first of all, to define and determine values of the quantities which are required for further calculations. Complexity of phenomena involved in the specific engagement of toothed rims in strain wave transmissions was taken into account in developing the calculation models, concerning:Engagement geometry (trajectories of the displacement of characteristic points and the resulting relative path of the flexspline tooth, position of the engagement point, and curvature radius of the engaged surfaces at that point);Engagement kinematics (speed distribution at the engagement point, average velocity of oil stream in the oil clearance);Distribution of contact forces and stresses in the engagement area.

The above-listed quantities depend, among other things, on the generator position angle *φ_G_*, and change for each value of the quantity. For that reason, a method was developed allowing determination of the parameters as functions of the angle *φ_G_*. The corresponding algorithm is presented in Figure 6.

### 2.3. Elasto-Hydrodynamic Oil Film in Meshing of Harmonic Drive Gears

The problem of elasto-hydrodynamic isothermal oil film in the meshing of harmonic drive gears can be described with the use of the following set of equations [16,17,18,34]:Pressure distribution in the spline engagement area:
(3)$$\frac{\partial }{\partial x}\left(\rho \;\times \;\frac{h^{3}\left(x\right)}{\eta \left(x\right)}\;\times \;\frac{\partial p}{\partial x}\right)=6\;\times \;\frac{\partial \left(\rho \;\times \;h\right)}{\partial x}$$
where:
*ρ = ρ*(*x*, *p*)(kg/m^3^)oil density,*h = h*(*x*)(m)oil film height,*p = p*(*x*)(Pa)distributed pressure in the circular spline-flexspline contact zone,*u*_0_(m/s)average oil stream velocity in the teeth clearance given by the formula [34]:
(4)$$u_{0}=\frac{u_{1}+u_{2}}{2}$$
*u*_1_(m/s)flexspline tooth velocity component tangent to tooth profile at the contact point,*u*_2_(m/s)circular spline tooth velocity component tangent to tooth profile at the contact point;

Stress in the flexspline tooth–circular spline tooth contact zone:(5)$$\sigma_{Hmax}=\frac{2F^{\prime }}{\pi a}$$
where:
*σ_Hmax_*(Pa)maximum normal stress,*F′*(N/m)the load force per the linear contact length given by the formula:
(6)$$F^{\prime }=\frac{F_{0}}{L}$$
*a*(m)contact area half-width given by the formula
(7)$$2a=\sqrt{\frac{8F^{\prime }R}{E^{\prime }}}$$
*E*′(Pa)reduced Young’s modulus given by the formula:
(8)$$\frac{1}{E^{\prime }}=\frac{1}{2}\left(\frac{1-\nu_{1}^{2}}{E_{1}}+\frac{1-\nu_{2}^{2}}{E_{2}}\right)$$
*R*(m)reduced curvature radius at the point of contact of a flexspline tooth profile with the circular spline tooth profile given by the formula:
(9)$$\frac{1}{R}=\frac{1}{r_{c1}}-\frac{1}{r_{c2}}$$
*L*(m)the length of linear contact between meshed flexspline and circular spline teeth,*E*_1_(Pa)Young’s modulus of flexspline material,*E*_2_(Pa)Young’s modulus of circular spline material,*ν*_1_(—)Poisson number of flexspline material,*ν*_2_(—)Poisson number of circular spline material,*r*_*c*1_(m)curvature radius at the contact point on flexspline tooth side,*r*_*c*2_(m)curvature radius at the contact point on circular spline tooth side;

Surface elastic deformation:(10)$$w\left(x\right)=-\frac{2\left(1-\nu_{1,2}^{2}\right)}{\pi E}\int_{-a}^{a}p\left(x\right)ln\left|x-s\right|ds$$
where:
*w = w(x)*(m)deformation,*s*(m)a variable determining the position of load application per unit surface area in the adopted system of coordinates;

Quantities characterizing lubricant properties:(11)$$\eta =\eta_{0}\times e^{\alpha \cdot p},\;\rho =\rho \left(p\right)$$
where:
*η*(Pa·s)dynamic viscosity of the lubricant,*η*_0_(Pa·s)dynamic viscosity at reference temperature,*α*(Pa^–1^)pressure viscosity coefficient.

Figure 7 shows a model of elasto-hydrodynamic oil film in a meshing of strain wave gear splines.

One of the quantities characterizing the properties of the oil film is the oil film minimum height. This is the least elasto-hydrodynamic film thickness which occurs at the lubrication gap constriction at the high-pressure zone end (Figure 7). The quantity is determined from the formula [17]:(12)$$h_{min}\;=\;1.6\alpha^{0.6}\;\times \;{\left(\eta_{0}\;\times \;u_{0}\right)}^{0.7}\;\times \;E^{\prime 0.03}\;\times \;R^{0.43}\;\times \;{\left(\frac{L}{F_{0}}\right)}^{0.13}$$
where *u*_0_ = (*u*_1_ + *u*_2_)/2 is the slip speed at the point of contact between the two engaging surfaces. The quantity is calculated as the difference of the tangent velocities *u*_1_ and *u*_2_ of the engaged surfaces, which are the flexspline tooth side and the circular spline tooth side, respectively. In view of the fact that in this analysis, the rigid circular spline is immobile (*u*_2_ = 0), so the slip speed will be:(13)$$u_{0}\;=\frac{u_{1}}{2}$$

As already mentioned, the oil film minimum height can be used as one of the criteria for assessment of the quality of lubrication provided to the meshed toothed rims. Engagement of teeth will be better, the lower the coefficient of friction is between the teeth side surfaces. To ensure that the gears are engaged with fluid friction, the oil film at the point of its minimum height should be sufficiently thick to separate the engaged surfaces completely. Geometrical structure of teeth side surfaces can be described by means of the roughness parameter Rz (μm) (Figure 8).

It is assumed that the condition of separation of the engaged surfaces is met when:(14)$$h_{min}\;\;\geq \;h_{adm}\;=\;1.1\left(Rz_{1}\;+\;Rz_{2}\right)$$

## 3. Results

Numerical simulations were carried out for a specific generator position determined by a generator position angle *φ_G_ =* 0.2063 rad. The adopted transmission operating parameters together with data concerning material properties of the components are listed in Table 4.

The tests were carried out for oils, parameters of which are presented in Table 5.

Results of application of the theoretical model to three oils with different viscosity values are presented in Figure 9, Figure 10 and Figure 11 in the form of oil film static characteristics.

As a result of analysis of the course of function *h_min_* (*n*_in_, *η*_0_, *F*_0_) shown in Figure 9, Figure 10 and Figure 11, the input shaft admissible speed *n_adm_* was determined for which the condition *h_min_ ≥ h_adm_* was met. In Figure 12, the effect of the oil viscosity *η*_0_ and the load *F*_0_ on the value of the speed *n_adm_* is presented.

## 4. Discussion

Based on numerical simulations performed with the use of a computer program, the following conclusions can be formulated:The oil film minimum height increases with increasing value of the product *η*_0_·*u*_0_;The increase in the reference viscosity value from *η*_0_ = 0.06 Pa·s to *η*_0_ = 0.295 Pa·s results in a 5.7-fold increase in the oil film minimum height (Figure 9);For the oil characterized with the reference viscosity coefficient *η*_0_ = 0.06 Pa·s, fluid friction occurs for the input shaft speed *n_in_* exceeding the value of 5670 rpm, whereas for the oil with the reference viscosity *η*_0_ = 0.295 Pa·s, fluid friction can be observed already for *n_in_* > 1194 rpm (Figure 9);With the decreasing value of force *F*_0_, the speed value at which the condition *h_min_*≥ *h_adm_* is met decreases accordingly. The most significant effect of value of the force *F*_0_ can be observed for the oil with VG68 viscosity grade;Value of the input shaft admissible speed *n_adm_* depends on, among other things: oil parameters (*α*, *η*_0_); the transmission load (*F*_0_/*L*); materials of which the toothed wheels were made (*E′*); and the teeth meshing geometry (*R*). It should be considered, however, that the average oil stream velocity *u*_0_, the force *F*_0_, and the reduced curvature radius *R* depend on the generator rotation angle *φ_G_*. That means that the speed *n_adm_* is also a function of the angle *φ_G_* and its value will be different at any working point;For transmissions operated at lower speeds and higher loads, in view of specific engagement of toothed wheels in harmonic drives, it is recommended to use oils with elevated viscosity. This follows from, among other things, the course of functions plotted in Figure 12;Selecting the lubricant for a harmonic drive, it is necessary to take into account, among other things, the reduction rate. Transmissions with high rates are characterized by the fact that the oil stream speed values in the cam flexible bearing will be much higher than those in the meshing. It is therefore necessary to select the lubricant type and parameters in a way enabling the formation of an oil film in the cam flexible bearing and in the meshing at the same time.

The obtained results of theoretical considerations and conclusions following from weight are in favor of using oils as lubricants in strain wave gears. It is worthwhile collating the obtained oil film characteristics with those recommended by the manufacturer HarmonicDrive (Table 1) [2]. The oils are characterized by viscosity, corresponding to VG68 grade. However, as can be seen from the plot shown in Figure 12, the use of oil with such viscosity enables generation of a film with the desired height only for *n_in_* speed in the range above about 4400 rpm for the load defined by the force in meshing *F* = 60 N and above 5700 rpm for the force *F* = 240 N. As shown in Table 2, the transmissions offered by the manufacturer [2], the parameters of which are close to those of the drive examined by us, can operate within this specific range of input shaft speeds. Some thought, however, should be given to the issue as to whether the oil with VG68 viscosity grade is suitable for transmissions operating at lower speeds *n_in_* (*n_in_* < 2000 rpm). It follows from the performed research that, in such cases, it is recommendable to use oils with higher viscosity, such as VG150 and above, especially in more heavily loaded transmissions. It should also be remembered that with increasing viscosity, properties of an oil become increasingly closer to those of a plastic grease, which triggers another problem—how to select oil parameters for the transmissions with large reduction ratios (*i_r_* > 150) in which the speed of the generator is much higher than that of the flexspline. It is, therefore, possible to assume that the transmission ratio is one of the limitations for increasing the oil viscosity.

The presented results concern a single operating point determined by the generator rotation angle *φ_G_*. For other values of the angle, quantities subject to changes include, among other thing, the contact geometry, the force in meshing, and the average oil stream velocity in tooth clearance. The quantities have an important effect on the oil film minimum height *h_min_*. This means that there are good reasons to investigate the film oil properties within the whole range of variability of the angle *φ_G_* because, according to the publication [35], the resultant velocities at points *M* and *N* (Figure 2) are increasing functions of the generator rotation angle. Admittedly, the referenced publications [12,32] present results of research on harmonic transmissions lubricated with oils; however, the studies did not include examination of the effect of the angle *φ_G_* on the parameters *F*, *u*_0_, and *R*. That creates a gap in the knowledge which should be filled by ways of further research. The model worked out by us will serve, in further stages of research, as a base for the development of software which will enable the determination of oil film properties within the full range of angle *φ_G_* and present the results in a clear and limpid way.

## Figures and Tables

**Figure 1 materials-14-01194-f001:**
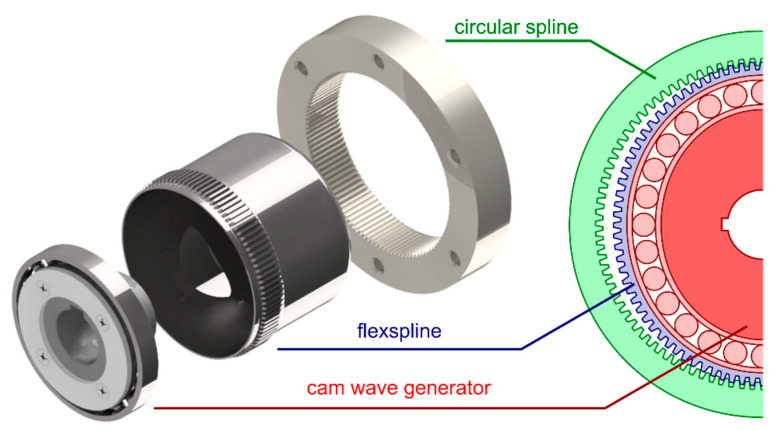
Intermeshing of flexspline with rigid circular spline in a harmonic drive with a double-wave elliptical cam generator.

**Figure 2 materials-14-01194-f002:**
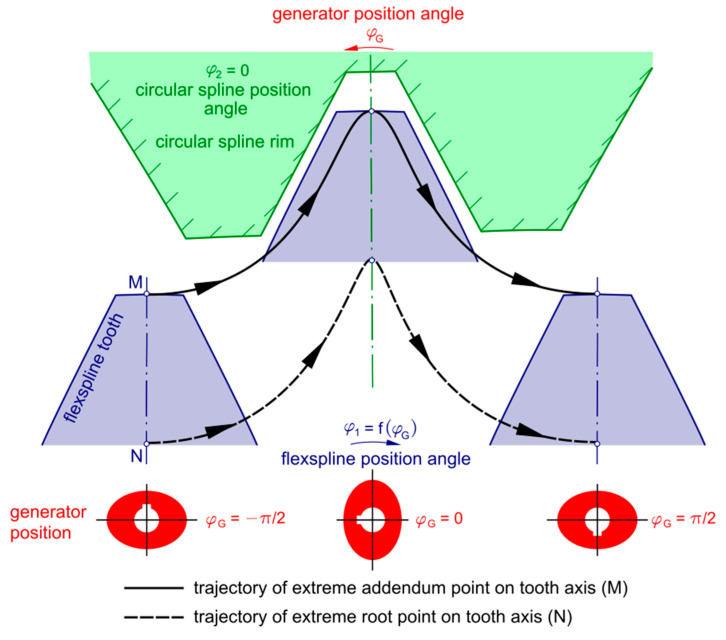
Trajectories of a flexspline tooth displacement relative to a circular spline tooth space, where *φ_G_* is the generator rotation angle.

**Figure 3 materials-14-01194-f003:**
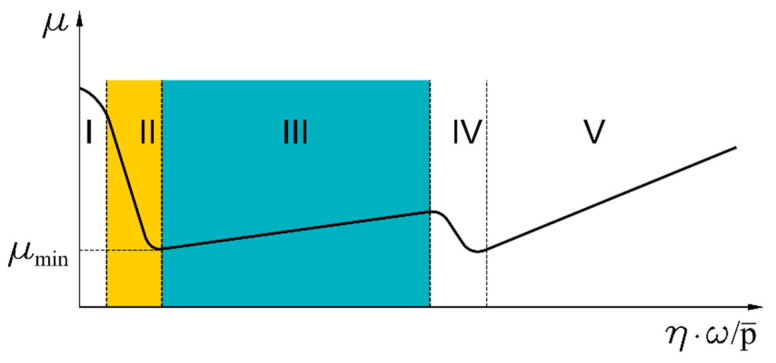
The Hersey curve, where: I—dry friction area; II—mixed friction area; III—fluid friction area; IV—transition from laminar to turbulent flow; V—increase in the coefficient of friction; *μ*—coefficient of friction; *μ_min_*—minimum value of the coefficient of friction; *η*—viscosity; *ω*—angular velocity; and $\bar{p}$
—surface loads [16].

**Figure 4 materials-14-01194-f004:**
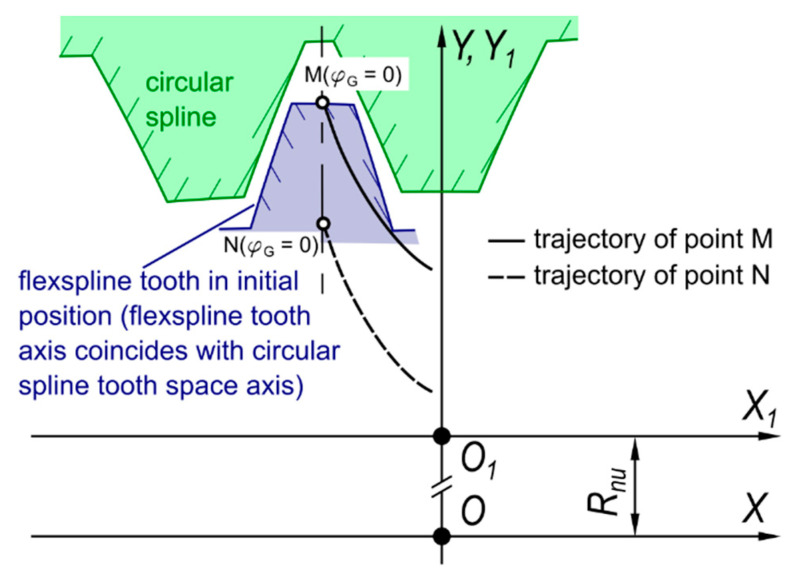
Reference systems *XOY* and *X*_1_*O*_1_*Y*_1_ adopted to describe the geometry and kinematics of harmonic drive meshing.

**Figure 5 materials-14-01194-f005:**
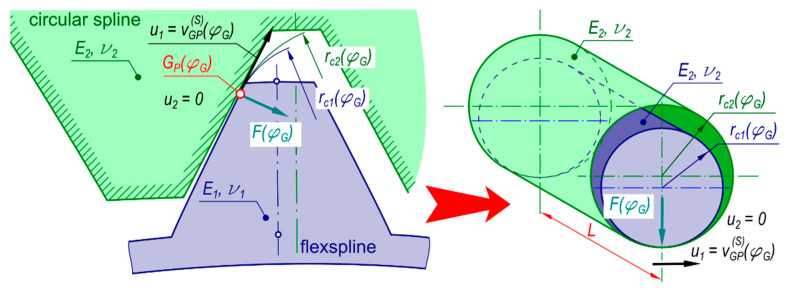
Construction of the substitution model for the contact of toothed rims in a strain wave transmission.

**Figure 6 materials-14-01194-f006:**
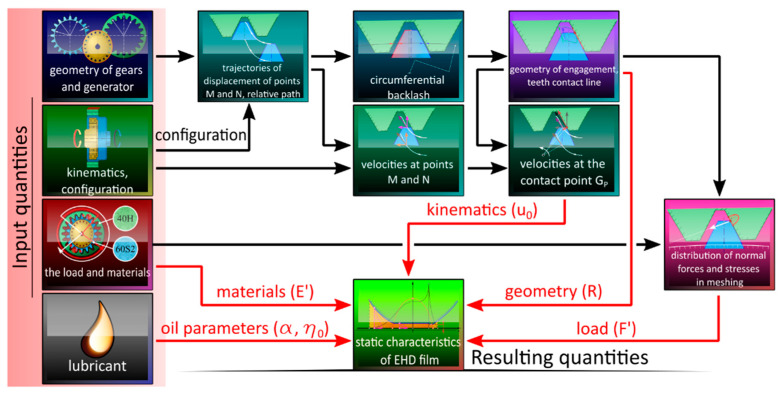
A simplified algorithm of the developed method.

**Figure 7 materials-14-01194-f007:**
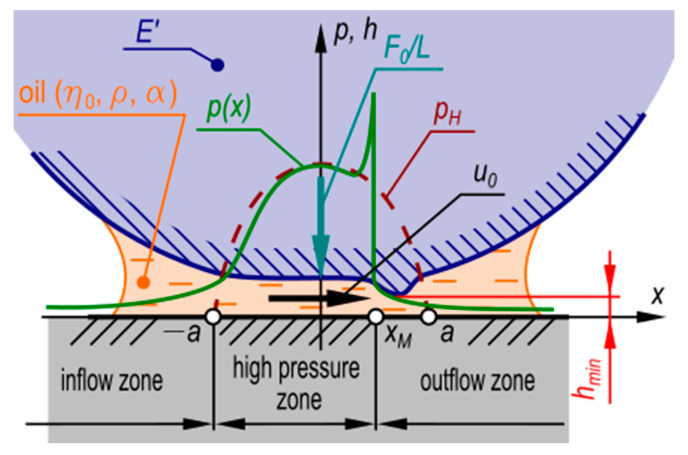
A model of elasto-hydrodynamic oil film: *η*_0_—dynamic viscosity; *α*—pressure viscosity coefficient; *E′*—reduced Young’s modulus; *p(x)*—pressure distribution curve; *p_H_*—Hertz contact pressure distribution curve; *h(x)*—oil film height; *h_min_*—oil film minimum height; *F*_0_—load force; *u*_0_—slip speed [16,17,18,34].

**Figure 8 materials-14-01194-f008:**
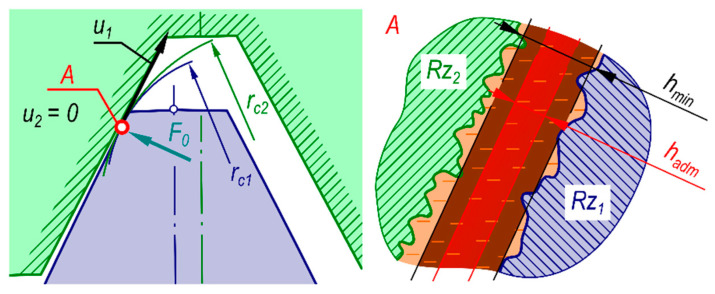
Engagement of a flexspline tooth side with a circular spline tooth side: *u*_0_—the slip speed; *u*_1_—flexspline tooth velocity component tangent to tooth profile at the contact point; *F*_0_—normal component of force at the contact point; *r*_*c*1,2_—curvature radii of flexspline and circular spline teeth sides, respectively, at the contact point; *Rz*_1,2_—roughness of flexspline and circular spline teeth side surfaces, respectively, expressed by the parameter Rz; *h_min_*—the minimum oil film height; *h_adm_*—the minimum admissible oil film height.

**Figure 9 materials-14-01194-f009:**
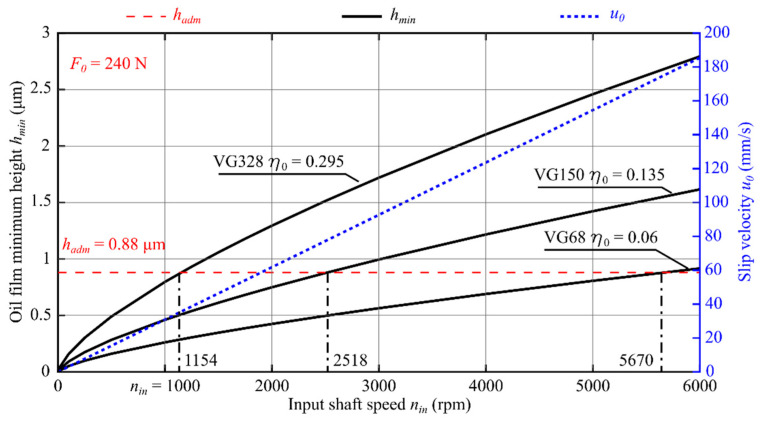
An oil film static characteristic at a given position *φ_G_ =* 0.2063 rad and load force in the meshing *F*_0_ = 240 N, where *n*_in_ (rpm) is the input shaft speed and *η*_0_ (Pa·s) is the oil dynamic viscosity.

**Figure 10 materials-14-01194-f010:**
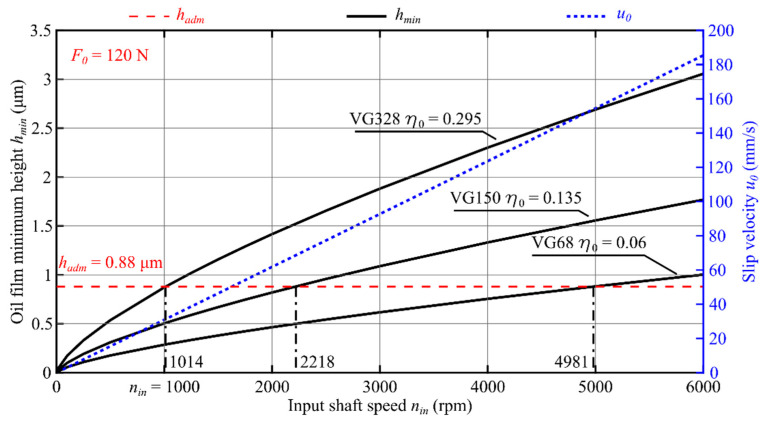
An oil film static characteristic at a given position *φ_G_ =* 0.2063 rad and load force in the meshing *F*_0_ = 120 N, where *n*_in_ (rpm) is the input shaft speed and *η*_0_ (Pa·s) is the oil dynamic viscosity.

**Figure 11 materials-14-01194-f011:**
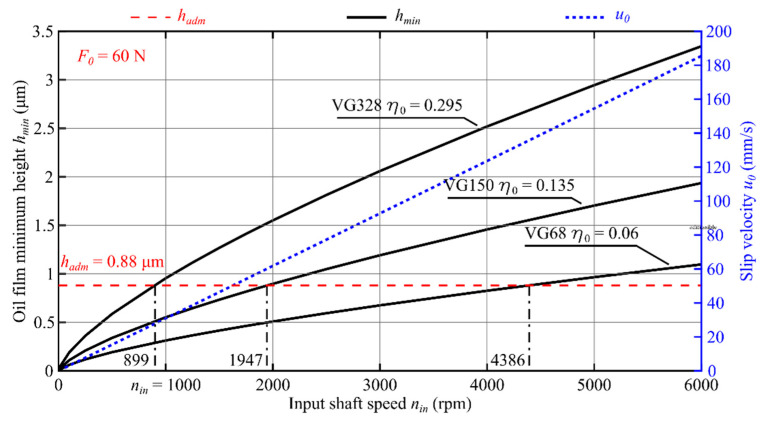
An oil film static characteristic at a given position *φ_G_ =* 0.2063 rad and load force in the meshing *F*_0_ = 60 N, where *n*_in_ (rpm) is the input shaft speed and *η*_0_ (Pa·s) is the oil dynamic viscosity.

**Figure 12 materials-14-01194-f012:**
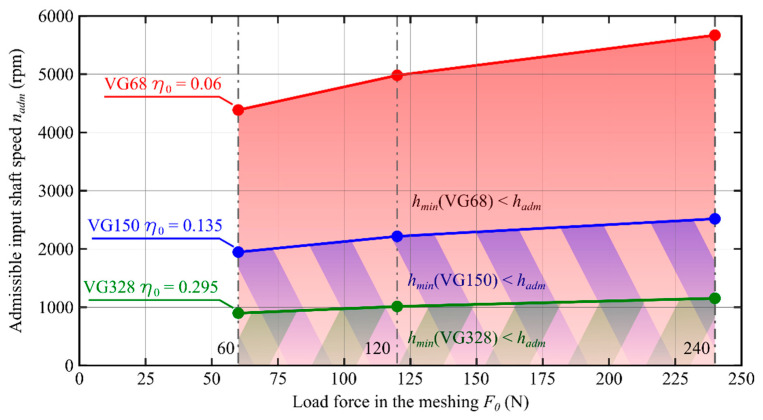
Admissible input shaft speed *n_adm_*.

**Table 1 materials-14-01194-t001:** Plastic greases used for strain wave gears.

Manufacturer	Grease	Operating Temperature Range
Harmonic Drive^®^	HarmonicGrasse^®^ SK-1A	0–40 °C
	HarmonicGrasse^®^ SK-2	0–40 °C
	HarmonicGrasse^®^ SK-3	0–40 °C
	HarmonicGrasse^®^ 4B No. 2	−10–70 °C
Laifual Drive	LF-II	−30–100 °C
	LF-III	−30–100 °C
	LF-IV	−30–100 °C

**Table 2 materials-14-01194-t002:** The relationship between the maximum admissible input shaft speed and the used lubricant in case of series CSF harmonic drives (HarmonicDrive^®^).

Size (-)	Reduction Ratio (-)	Torque Capacity at 2000 rpm (N·m)	Maximum Input Speed (rpm)		Average Input Speed (rpm)	
			Oil	Plastic Grease	Oil	Plastic Grease
Series CSF						
8	100	2.4	14,000	8500	6500	3500
20	30	15	10,000	7300	6500	3500
45	120	402	5000	3800	3300	3000
90	50	1180	2700	2000	2100	1300
100	160	3550	2500	1800	2000	1200

**Table 3 materials-14-01194-t003:** Oils used in harmonic drives.

Oils			
Manufacturer	Grade	Manufacturer	Grade
Class-2 standard transmission oil (for very high pressure applications)	ISO VG68	Japan Energy	ES gear G68
Mobil Oil	Mobilgear 600XP68	NIPPON Oil	Bonock M68 Bonock AX68
Exxon	Spartan EP68	Idemitsu Kosan	Daphne super gear LW68
Shell	Omala Oil 68	General Oil	General Oil SP gear roll 68
COSMO Oil	Cosmo gear 68	Klüber	Syntheso D-68EP

**Table 4 materials-14-01194-t004:** Adopted parameter values.

Parameter	Symbol	Value
Input shaft rotational speed	*n* _in_	1000 rpm
Input shaft angular speed	*ω* _in_	104.7198 rad/s
Generator rotation angle	*φ_G_*	0.2063 rad/11.82 deg
Flexspline tooth velocity at the contact point, tangent to profile	*u* _1_	60.1193 mm/s
Load force in the meshing (assumed based on the curve) [34]	*F* _0_	240 N
Reduced curvature radius at the contact point (for concave surface curvature radius *r*_c1_ = 19.2194 mm and convex surface curvature radius contact *r*_c2_ = 19.2307 mm—Figure 5)	*R*	32.585 m
Reduced Young’s modulus (60S2 steel/40H steel)	*E*′	358.7 GPa
Roughness of flexspline tooth side surface	*Rz* _1_	0.4 µm
Roughness of circular spline tooth side surface	*Rz* _2_	0.4 µm
**Flexspline Rim Parameters**		
Reference circle radius	*r* _1_	39.6 mm
Root circle radius	*r* _f1_	40.824 mm
Addendum circle radius	*r* _a1_	41.858 mm
Addendum modification coefficient	*x* _1_	3.39
Flexspline body maximum radial distortion	*w* _0_	0.64 mm
**Circular Spline Rim Parameters**		
Pitch circle radius	*r* _2_	40.2 mm
Root circle radius	*r* _f2_	42.7681 mm
Addendum circle radius	*r* _a2_	41.658 mm
Addendum modification coefficient	*x* _2_	3.55
Circular spline rim facewidth	*L*	12 mm

**Table 5 materials-14-01194-t005:** Parameters of oils subjected to testing.

Parameter	Symbol	Value
Pressure viscosity coefficient	*α*	0.02·10^−6^ Pa^−1^
Dynamic viscosity	*η* _0_	0.06 Pa·s/0.295 Pa·s
Reference temperatrure	*T* _0_	40 °C

## Data Availability

The data presented in this study are available on request from the corresponding author.

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
