# Peer review of "Properties of Elasto-Hydrodynamic Oil Film in Meshing of Harmonic Drive Gears"

_materials, 2021, doi:10.3390/ma14051194_

Round 1
Reviewer 1 Report
Althougth the authors reported a priori manuscript dealing with an interesting issue, such as the modelling of the lubricating material used in harmonic drive gears, this manuscripts contains just one or two paragraphs, along with one graph of results. The rest of the manuscript is just a more than reasonable explanation of the methodology followed during the modelling. Therefor, I esteem the revealed results quite poor to be published in the present form. As a consequence, even though I can see the hard work behind those limited results and discussion lines, I would recommend either a deeper evaluation of the obtained results or include in this manuscript what they indicated in lines 374-375 as the next research step “The next stage of the research will be focused on determination of the effect of geometry and load on the oil film minimum height.”.
Reviewer 2 Report
The present paper is focused on the elastohydrodynamic lubrication of drive gears. Although the topic is definitely worth investigation, the manuscript in the present form is unfortunately not suitable for publication in the Materials journal.
- The structure of the paper (especially the Introduction part) is very chaotic. I miss deeper motivation, clear goal of the study. A lot of subchapters and the tables including a lot of parameters make it even more difficult to follow the main thoughts of the manuscript.
- English needs to be considerably improved.
- Although the background about the drives and designs is useful, the Introduction part needs to be considerably improved by including other studies dealing with EHD lubrication of gears and drives. Studies [21]-[30] seem to be appropriate but these need to be discussed in further detail showing some implications for the present work.
- The formatting of equations seems to be incorrect.
- 4 and Tab. 5 do not belong to the Results section (it belongs to M&M).
- The results section is very short – NOT APPLICABLE.
- There is NO discussion included – this is a mandatory requirement to the authors. Without the discussion of the obtained results with other research works, there is no way to publish the paper in the peer-reviewed journal with impact factor. – NOT APPLICABLE.
- Concluding remarks are very weak without any clear implication for practice or gear design.
Reviewer 3 Report
- The authors have shown that when using the more viscous oil, the critical thickness of the lubricating layer is achieved at a lower shaft speed. This conclusion is more than obvious and hardly represents new information. However, in the simulation it was assumed that the pressure viscosity coefficient has the same value for oils with different viscosities. This is a very conditional assumption, since the pressure viscosity coefficient of the more viscous oil is usually higher than that of a less viscous one. Moreover, the article provides information on the variety of oils used in harmonic drives. I believe that the authors could include in the article the results of calculations for different values of the pressure viscosity coefficient.
- The sum of the roughness parameters 1.1 * (Rz1 + Rz2) is taken as the critical thickness of the lubricating layer. In this case, the values Rz1 = Rz2 = 0.4 μm. However, such values of the Rz parameter are typical for the 11th grade of purity. And in the contact hydrodynamic theory, it is generally accepted that the sum (Ra1 ^ 2 + Ra2 ^ 2) ^ 0.5 is taken as the critical thickness of the lubricating layer for parts with a surface finish of 11 and higher. When the value of the parameter Rz = 0.4 μm, the value of the parameter Ra = 0.08 μm should be expected. In this case, the critical thickness of the lubricating layer is about 0.113 μm. This greatly affects the interpretation of the calculation results. The authors need to justify the value of the thickness of the lubricating layer, which they take as critical.
- Obviously, oils for harmonic drives contain anti-wear additives, which form an adsorbed boundary layer on the friction surfaces with their specific properties (elasticity, viscosity increase as the surface is approached). The size of this layer is comparable to the height of microroughnesses and can exceed the value of Rz. This leads to the antiwear properties of the oils. The authors should have indicated in the text of the article the assumption that the effect of antiwear additives on the minimum thickness of the lubricating layer is not taken into account.
Round 2
Reviewer 1 Report
The authors have increased the result content of the article. I still consider this study a very interesting issue. Given that this research deals with a numerical simulation, I would recommend to include the limits until which the results may be regarded as "correct".
Moreover, I would recommend to reorganize the discussion section, thus exhibiting the conclusion but avoiding the list of points the authors have included.
Reviewer 2 Report
The manuscript was considerably improved. However, it still lacks discussion. The Discussion part in the revised version is not discussion; it is a Conclusion. Every peer-reviewed paper should include original discussion where the authors confront their findings with literature, discuss the limitations and maybe highlight some implications for practice/future research. In my humble opinion, the paper without appropriate discussion should not be published.
